# Long-term relative survival in uveal melanoma: a systematic review and meta-analysis

Gustav Stålhammar [1,2✉] & Christina Herrspiegel[1,2]

## Abstract

**Background** A large proportion of patients with uveal melanoma develop metastases and succumb to their disease. Reports on the size of this proportion vary considerably.

**Methods** PubMed, Web of Science and Embase were searched for articles published after 1980. Studies with ≥100 patients reporting ≥five-year relative survival rates were included. Studies solely reporting Kaplan-Meier estimates and cumulative incidences were not considered, due to risk for competing risk bias and classification errors. A meta-analysis was performed using random-effects and weighted averages models, as well as a combined estimate based on curve fitting.

**Results** Nine studies and a total of 18 495 patients are included. Overall, the risk of selective reporting bias is low. Relative survival rates vary across the population of studies ($I^2$ 48 to 97% and $Q$ $p < 0.00001$ to 0.15), likely due to differences in baseline characteristics and the large number of patients included ($\tau^2 < 0.02$). The 30-year relative survival rates follow a cubic curve that is well fitted to data from the random-effects inverse-variance and weighted average models ($R^2 = 0.95$, $p = 7.19E^{-7}$). The estimated five, ten, 15, 20, 25 and 30-year relative survival rates are 79, 66, 60, 60, 62 and 67%, respectively.

**Conclusions** The findings suggest that about two in five of all patients with uveal melanoma ultimately succumb to their disease. This indicates a slightly better prognosis than what is often assumed, and that patients surviving 20 years or longer may have a survival advantage to individuals of the same sex and age from the general population.

### Plain language summary

Relative survival compares how persons with and without a disease survive. It is a good way to describe the chances of surviving uveal melanoma, which is the most common cancer inside the eyes of adults. In this analysis, information from several studies were collected to make an estimation of the relative survival of patients with uveal melanoma. We find that about two in five patients will die from this disease over very long time. This is slightly better than what has often been reported before. In fact, patients that survive for more than 20 years may even fare better than persons without the disease. These findings are useful when counselling patients and relatives about the chances of surviving uveal melanoma.

[1] St. Erik Eye Hospital, Box 4078, Solna, 171 04 Stockholm, Sweden. [2] Department of Clinical Neuroscience, Karolinska Institutet, Tomtebodavägen 18A, plan 5, 171 77 Stockholm, Sweden. ✉email: gustav.stalhammar@ki.se

**M**elanomas of the uvea are the most common primary intraocular malignant tumors in adults, affecting more than 7000 individuals each year worldwide[1]. No substantial survival differences have been observed between commonly used treatment modalities, patient sex and age or calendar period during the last several decades[2–6]. Eventually, a large proportion of patients develop distant metastases after which median survival is about one year[7]. Currently available treatment options for primary tumors have limited effect on patient survival[3]. Similarly, there are no clinically available treatments with meaningful impact on survival in metastatic disease[8].

Estimations of the proportion of patients that develop metastases vary considerably. It is often stated that one half of patients will die from their disease[9–14]. Other publications report significantly lower mortality rates in the range of 20–25% for patients with similar patient baseline characteristics over similar periods of time[2,15,16]. The reasons for this variance may at least partially be found in dissimilarities in the methods used for calculation of the mortality rates. No method of estimation of the long-term mortality in disease is without flaws, but some methods of estimation of mortality rates may be less suitable than others. Actuarial methods including life tables and Kaplan–Meier estimates are excellent for evaluation of all-cause mortality, but are likely to overestimate disease-specific mortality in the presence of competing risks, i.e., death from other causes[17–19]. Cumulative incidences of melanoma-related mortality rely on accurate classifications of the cause of death. This has relatively small impact when studying diseases with low mortality, but may be more biased in studies of a disease with a mortality that approaches 50%[20]. A previous study found that uveal melanoma-related death was misclassified in more than half of cases included in a national cause of death registry[21]. Relative survival, in which the observed overall survival of a cancer population is divided by the overall survival in a reference population without the cancer is less prone to bias provided that the sample size is sufficiently large, that the disease is rare in the general population and that it does not have risk factors that are strongly associated with other causes of death (e.g., smoking)[20]. Analysis of relative survival may therefore be well suited for uveal melanomas.

What do we answer patients that ask us the basic question about how high the mortality is in their disease? Patients with choroidal or ciliary body melanomas rarely undergo biopsy or other tumor sampling prior to treatment and before we have results from radiological and detailed ophthalmological examinations, an individualized prognosis is not available. This study is intended to help us inform patients and relatives with a systematic review and meta-analysis of long-term relative survival rates. Based on nine included studies and a total of 18,495 patients, we estimate 5, 10, 15, 20, 25, and 30-year relative survival rates of 79, 66, 60, 60, 62, and 67%, respectively. This indicates a slightly better prognosis than what is often assumed, and that patients surviving 20 years or longer may have a survival advantage to individuals of the same sex and age from the general population.

## Methods

**Search strategy and selection criteria**. We did a meta-analysis to evaluate the long-term relative survival in uveal melanoma. Data was acquired with a comprehensive literature search in the PubMed, Web of Science and Embase databases for peer reviewed published articles that described relevant results. The following search terms were used and matched to appropriate medical subject headings: ("uveal melanoma" OR "choroidal melanoma" OR "ciliary body melanoma") AND "relative survival". The search strategy was restricted to titles and/or abstracts of human

clinical studies published after January 1st 1980 in English or any language for which an English translation was readily available[22]. The latest search was performed on August 19, 2021. All available studies were included and could be accessed in full via the University Library, Karolinska Institutet. Study authors were contacted if discrepancies existed, for clarifications or if we thought that additional unpublished data could be useful for this analysis. Trial registries, unpublished studies, gray literature, animal studies, laboratory studies, letters to the editor, correspondence, notes, editorials, and conference abstracts were not considered. Reference lists of included articles were searched for additional studies. As both clinical trials and observational studies could be considered, the search method was based on the guidelines of the Preferred Reporting Items for Systematic Reviews and Meta-analyses (PRISMA) and on the checklist for Meta-analyses Of Observational Studies in Epidemiology (MOOSE)[21,23]. The PRISMA and MOOSE checklists are available as related manuscript files. The protocol was registered and published in advance on PROSPERO (CRD42021265504).

The selection of articles for this analysis was performed in four steps: identification, abstract and full-text screening, eligibility assessment, and inclusion. Abstract screening of articles identified in the literature search was done independently by the two authors, with any disagreements resolved by discussion. Publications were included for full-text screening if they reported (1) ≥5-year relative survival rates (or data that could be readily converted to relative survival) for patients with uveal melanoma, (2) consecutively or prospectively included patients, (3) at least 100 patients. Studies were excluded if they (1) only reported survival rates for prognostically relevant subgroups (e.g., tumors with specific mutations, gene expression profiles, histological appearance, or size categories), (2) were earlier versions of a series of articles from the same database or center, (3) reported patients that were already included in another publication, or (4) did not provide confidence intervals or standard errors for their relative survival estimates. Studies including patients with primary conjunctival or orbital melanomas or metastatic lesions were not considered. The same inclusion and exclusion criteria applied to full-text screening (if not evident in title or abstract). Additionally, articles could be excluded if they were deemed to have sub-par methodological quality, as described below.

**Quality assessment of studies**. All articles that reached the eligibility assessment step was evaluated with a modified version of the Newcastle-Ottawa Scale-Education (NOS-E)[24]. At the eligibility assessment step, no article was excluded because it was deemed to have sub-par methodological quality according to NOS-E.

**Data collection, qualification of searchers and risk for bias assessment**. Relative survival rates were extracted from downloaded full texts of each included study. The data was not coded. Dr. Herrspiegel is an ophthalmologist and ocular oncology and pathology researcher. Dr. Stålhammar is a board-certified ophthalmologist and pathologist, and his qualifications include a research group leadership of Ocular Oncology and Pathology at Karolinska Institutet, Stockholm, Sweden. Risk of bias was evaluated according to the recommendations of the Cochrane collaboration[22].

**Statistical analysis**. The meta-analysis was based on three pre-specified methods for calculating relative survival rates. The a priori determined outcome measure was the long-term relative survival rate, reported in 5-year intervals. The variance of survival rates across the population of studies was evaluated with $\tau^2$,

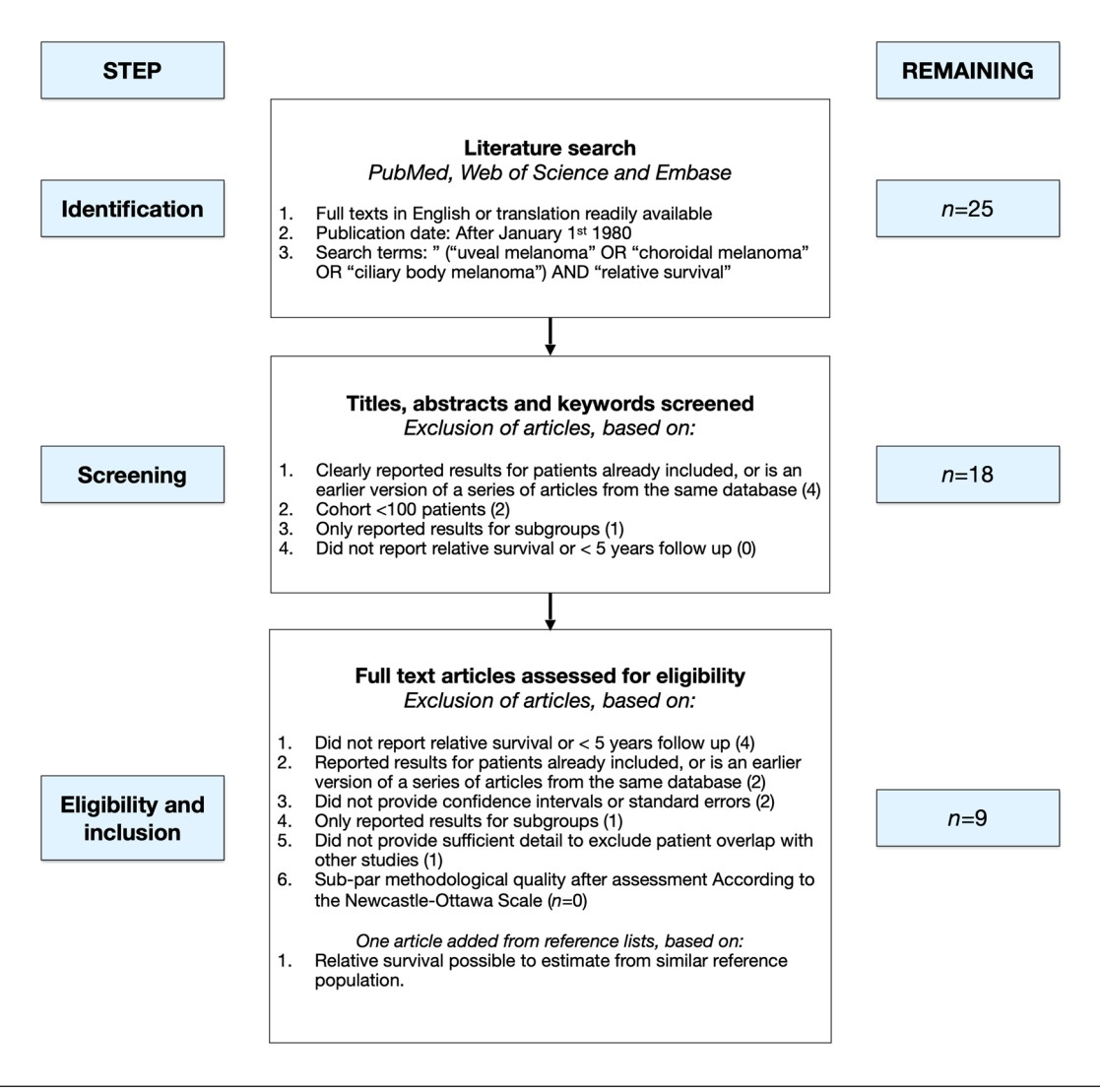

**Fig. 1 Flow diagram of study selection process.** The selection of articles for analysis was performed in four steps: identification, abstract and full-text screening, eligibility assessment, and inclusion.

which reflects the amount of true heterogeneity regardless of number of included studies or sample size[25]. The alternative measurements of heterogeneity $Q$ and $I^2$ were included for comparison[25,26]. Differences with a $p < 0.05$ were considered significant, all $p$ values being two-sided. When derived from cumulative overall survival, the 95% CI of the relative survival rates was calculated by dividing the standard error of the observed cumulative survival rate by the expected survival rate, which is a common method in cancer epidemiology (Eq. 1):[27–29]

$$95\% \, \text{CI} = \text{Relative survival rate} \pm 1.96 \frac{\text{standard error of the observed survival rate}}{\text{Expected survival rate}} \tag{1}$$

Equation (1): 95% confidence interval (CI) of a relative survival rate.

Firstly, the random-effects inverse-variance statistical method was used, with 95% CI. The weight given to each study was the inverse of the variance of the relative survival rate. To obtain the standard error from stated 95% CIs, the latter was divided

by 3.92 (Eq. 2).

$$\text{Standard error} = (\text{upper} - \text{lower limit of } 95\% \, \text{CI})/3.92 \tag{2}$$

Equation (2): Standard error calculated from 95% CI.

Thus, larger studies which have smaller standard errors were given more weight than smaller studies, which have larger standard errors. This weighting method minimizes the imprecision of the pooled effect estimate[22]. This statistical analysis was performed using the Review Manager by the Cochrane Collaboration (RevMan Version 5.4. Copenhagen: The Nordic Cochrane Center; The Cochrane Collaboration, 2014).

Secondly, weighted averages were calculated based on a model previously used for meta-analysis of shorter-term mortality in uveal melanoma[30]. The number of deaths at each point in each study, $n_\dagger$, was multiplied with the same study's sample size, $n$. The resulting product was divided by the total sum of weights $\Sigma n_\dagger n$ to arrive at a weight for each individual study. $\Sigma n_\dagger n$ was then multiplied by $n_\dagger/n$, resulting in a weighted average for each study, $p$, which was then summed for a pooled weighted average

**Table 1 Summary of primary studies included in the meta-analysis. Y, year(s).**

Relative survival

| # | Authors (ref.) | n | Country | Period of diagnosis | Tumor size | Group | 5 years (%) | 95% CI | 10 years (%) | 95% CI | 15 years (%) | 95% CI | 20 years (%) | 95% CI | 25 years (%) | 95% CI | 30 years (%) | 95% CI |
|---|---|---|---|---|---|---|---|---|---|---|---|---|---|---|---|---|---|---|
| 1 | Kujala et al.[17] | 289 | Finland | 1962-1981 | Mean T 7 mm, mean ø 13 mm | | 63 | 60-66 | 57 | 54-61 | 54 | 50-58 | 50 | 45-55 | 62 | 54-69 | 82 | 71-92 |
| 2 | Isager et al.[34] | 2178 | Denmark | 1943-1997 | N/a | Men | 66 | 62-69 | 55 | 51-60 | 49 | 44-54 | | | | | | |
| | | | | | | Women | 69 | 66-73 | 57 | 52-61 | 54 | 48-59 | | | | | | |
| 3 | Burr et al.[35] | 2876 | England and Wales | 1986-1999 | N/a | | 72 | 70-75 | | | | | | | | | | |
| 4 | Virgili et al.[36] | 1081 | Western and Eastern Europe* | 1983-1994 | N/a | Western Europe | 63 | 58-69 | | | | | | | | | | |
| | | | | | | Eastern Europe | 65 | 61-70 | | | | | | | | | | |
| 5 | Caminal et al.[39] | 155 | Spain | 1995-2004 | Small 13%, Medium 58%, Large 34%** | | 90 | 84-96 | | | | | | | | | | |
| 6 | Chew et al.[42] | 308 | Australia | 1981-2005 | ø ≤10 mm 22%, >10 mm 26%, unknown 52%. T, ≤5 mm 19, >5 mm 26, unknown 55% | | 81 | 76-87 | 71 | 63-78 | | | | | | | | |
| 7 | Baily et al.[16] | 253 | Ireland | 2010-2015 | Mean T 6.3 mm, mean ø 12.6 mm | | 77 | 68-84 | | | | | | | | | | |
| 8 | Stålhammar et al.[23] | 677 | Sweden | 1980-1999 | Mean T 5.2 mm, mean ø 10.1 mm | | 79 | 76-82 | 71 | 66-75 | 66 | 60-72 | 64 | 55-73 | 74 | 60-87 | 86 | 58-115 |
| 9 | Radivoyevitch et al.[49] | 10,678 | United States | 1975-2016 | N/a | | 81 | 80-82 | 70 | 69-72 | 64 | 63-66 | 60 | 57-63 | 60 | 56-64 | 61 | 56-67 |
| | Sum | 18,495 | | | | | | | | | | | | | | | | |

T thickness, ø diameter, N/a not available, y years
*Only patients from non-overlapping geographical regions included.
**Tumors classified as small if their apical thickness were between 1 and 3 mm and their longest basal diameter at least 5 mm, as medium if their apical thickness were between 3.1 and 10 mm and their longest basal diameter no more than 16 mm, and as large if their apical thickness were greater than 10.1 mm and/or their basal diameter greater than 16 mm.

(Eq. 3). 95% CI was calculated from the standard deviation of Σp.

$$\Sigma p = \frac{n_\dagger{}^2}{\Sigma n_\dagger n} \tag{3}$$

Equation (3): Pooled weighted average (Σp). $n_\dagger$, number of deaths at each point in each included study. $n$, study sample size. $\Sigma n_\dagger n$, total sum of weights in all included studies.

Thirdly and last, curve fitting was performed based on the results of the random-effects and weighted averages models to arrive at a combined estimate. 95% CI for the combined estimate was calculated by dividing the standard deviation of the random-effects and weighted averages models by the square root of the number of samples ($n = 2$) to obtain a standard error from which the 95% CI was derived (±1.96 × standard error). This statistical analysis was performed using SPSS statistics version 26 (IBM, Armonk, NY, USA).

**Role of the funding source**. The funders of the study had no role in study design, data collection, data analysis, data interpretation, or writing of the report. CH and GS had full access to all the data in the study and GS had final responsibility for the decision to submit for publication.

**Reporting summary**. Further information on research design is available in the Nature Research Reporting Summary linked to this article.

## Results

**Study selection**. The literature search resulted in 25 nonduplicate articles[5,6,16,21,23,31–50]. Seven of these did not reach the inclusion criteria upon abstract screening[6,37,40,41,43,45,50]. The full text of the remaining 18 articles were reviewed. Four studies were excluded since they did not report relative survival, did not present data that could be readily converted to measurements of relative survival and/or had less than 5 years of follow-up (not evident in title or abstract)[32,44,46,47]. Two were earlier versions of a series of articles from the same database or center[33,48]. Two articles did not report confidence intervals or standard errors for their relative survival estimates and could therefore not be used for statistical comparisons[21,31]. One study was excluded because it only reported survival for subgroups[5]. Another publication was based on data from 76 European cancer registries, but no details were provided to enable reliable assessment of overlap of patients included in other studies, why the study was excluded[38].

One frequently referenced publication from Finland reported the long-term cumulative incidence of disease-specific and overall survival[17]. The Nordic countries are similar in terms of economy, welfare and health care as well as in life expectancy and distribution of overall disease burden, smoking, and metabolic risk factors. Therefore, data on the remaining life expectancy in a Swedish reference population from the same time period could be used to estimate relative survival and 95% CI in the Finnish sample (Supplementary Table 1)[23,51]. One large study was based on data from 32 European cancer registries, of which three had been partially used in other included publications[21,34–36]. Therefore, only patients from the non-overlapping geographical regions (western and eastern European areas) were included. Lastly, in another publication, a separate cohort of 257 patients from an earlier calendar period had been included for comparison of the outcomes examined in the main cohort of the study, why only the main cohort was included in this meta-analysis.

Nine studies were eligible and included in the meta-analysis (Fig. 1)[16,17,23,34–36,39,42,49]. These represented a total of 18,495 patients.

**Table 2 Pooled estimates of relative survival in five-year intervals after uveal melanoma diagnosis in random effects and weighted averages models.**

|  | 5 years | | 10 years | | 15 years | | 20 years | | 25 years | | 30 years | |
|---|---|---|---|---|---|---|---|---|---|---|---|---|
|  | RS (%) | 95% CI | RS (%) | 95% CI | RS (%) | 95% CI | RS (%) | 95% CI | RS (%) | 95% CI | RS (%) | 95% CI |
| Random effects estimate | 75 | 70–80 | 65 | 58–72 | 59 | 52–66 | 58 | 50–65 | 63 | 57–69 | 74 | 56–92 |
| Weighted average | 79 | 76–82 | 69 | 63–75 | 63 | 55–72 | 60 | 49–71 | 60 | 49–72 | 61 | 50–72 |

*RS* relative survival.

**Characteristics of included studies**. Of the nine included studies, one included American patients, one Australian, one Swedish, one Danish, one Finnish, one English and Welsh, one Irish, one Spanish and one western and eastern European patients including German, Italian, Polish, Slovakian, and Slovenian but excluding countries and territories overlapping the other studies. The largest study reported 10,678 patients and the smallest 155 (Table 1). All studies were published between 2003 and 2021, and all studies were retrospective cohort studies. The included patients had been treated with enucleation, plaque brachytherapy, proton beam radiotherapy, transscleral local resection, endoresection, or transpupillary thermotherapy. The estimated one minus relative survival in the Finnish study did not differ more than 3% points from its reported cumulative incidence of melanoma-related mortality at any 5-year interval up until 20 years after prognosis[17]. With longer follow-up, survival rates diverged with increasing relative survival.

A majority of studies relied on mortality data from a combination of population-based registries including cancer registries and cause of death registries and the authors' institution own clinical records (Supplementary Table 2).

**Random-effects model**. The inverse-variance pooled estimate of the relative survival was 75% at 5 years after diagnosis (95% CI 70–80%), 65% at 10 years (58–72), 59% at 15 years (52–66), 58% at 20 years (50–65), 63% at 25 years (57–69), and 74% at 30 years (56–92, Table 2 and Fig. 2).

**Weighted averages model**. In calculation of weighted averages $\hat{\Sigma p}$, the pooled estimate of relative survival was 79% at 5 years after diagnosis (95% CI 76–82%), 69% at 10 years (63–75), 63% at 15 years (55–72), 60% at 20 years (49–71), 60% at 25 years (49–72), and 61% at 30 years (50–72, Table 2). Consequently, this method yielded slightly higher survival estimates during the first 20 years after diagnosis. The mean difference at each point in time between the random effects and weighted averages model was 1.1% points, which was not significant in non-parametric testing (Mann–Whitney $U$ $p = 0.94$).

**Combined estimate**. Goodness of fit was tested in model summaries and ANOVA tables. A cubic curve was best fitted to the data ($F$-score 65.3, $R^2 = 0.95$, $p = 7.19\text{E}^{-7}$). The relative survival rate as a function of time could be described as: $y = -0.002^3 + 0.193x^2 - 5.054x + 99.381$ where $y$ is the relative survival and $x$ is the year after diagnosis. The resulting estimate of relative survival was 79% at 5 years after diagnosis (95% CI 73–88%), 66% at 10 years (61–71), 60% at 15 years (55–66), 60% at 20 years (54–65), 62% at 25 years (57–68), and 67% at 30 years (50–85, Supplementary Table 3).

The pooled relative survival rates are illustrated for each individual study (Fig. 3a), for the random-effects model (Fig. 3b), for the weighted averages model (Fig. 3c), and for the combined estimate (Fig. 3d).

**Heterogeneity and risk of bias**. Overall, the risk of selective reporting bias was low according to the guidelines from the Cochrane collaboration: All included studies either had both study protocols available and had prespecified outcomes, or convincingly reported all prespecified outcomes in absence of an available protocol[22]. Instead, the main source for variances in the current meta-analysis may hypothetically have been differences in patient and tumor characteristics (clinical baseline heterogeneity), competing risk bias in measurements of very long-term outcomes (statistical heterogeneity) and differences in methods for classification of outcomes, e.g., in how causes of death were established and recorded (other sources of heterogeneity)[18,25]. There was indeed considerable variance across the population of studies ($I^2$ 48–97% and $Q$ $p < 0.00001$ to 0.15). As indicated by statistics not affected by sample size however, the main source for this heterogeneity was the large number of patients included ($\tau^2 < 0.02$, supplementary Table 4).

**Discussion**

In the present meta-analysis' combined estimate, patients have a relative survival of 60% at 15–20 years after uveal melanoma diagnosis. This indicates that about two in five of all patients will succumb to their disease within this time frame. Previous publications confirm that death from metastatic disease occur rarely after 20 years[17]. Interestingly, pooled relative survival rates suggest that patients surviving 20 years or longer may even have a survival advantage to individuals of the same sex and age from the general population. The causality behind this observation is beyond the scope of this meta-analysis. Hypothetically, the u-shaped relative survival curve may be explained by an accumulation of other risk factors (e.g., cardiovascular morbidity, smoking, and obesity) in patients dying from uveal melanoma, and a reduced presence of such risk factors among survivors.

The reported mortality rates in individual studies varied, even in cohorts with seemingly similar baseline patient characteristics. Regardless, only the widest confidence interval, produced by weighted estimates of the nine included studies, reached the 50% mortality that is so often quoted as a hallmark of uveal melanoma. This may be useful when counseling patients. Many authors stating a more pessimistic prognosis refer to the same excellent publication by Kujala et al.[17], in which the melanoma-related mortality for 289 included patients was 31, 45, 49, and 52% by 5, 15, 25, and 35 years after primary tumor treatment, respectively. However, this cohort only included patients that had undergone enucleation or exenteration, with relatively large tumors: The median tumor thickness was seven mm (range 1–20) and the median largest basal diameter (LBD) was 13 mm (range 3–25). These dimensions are slightly larger than the mean tumor in other published cohorts with long follow-up and better survival rates[23,52]. And, as shown by Shields et al., the risk for metastasis increases with each increased millimeter of tumor thickness[52]. The mortality rates presented by Kujala et al. coincide at least roughly with metastatic rates of seven mm thick posterior uveal melanomas (21 and 41% at 5 and 10 years, respectively) published by Shields et al.[52]. Similarly, they align well with the melanoma-related

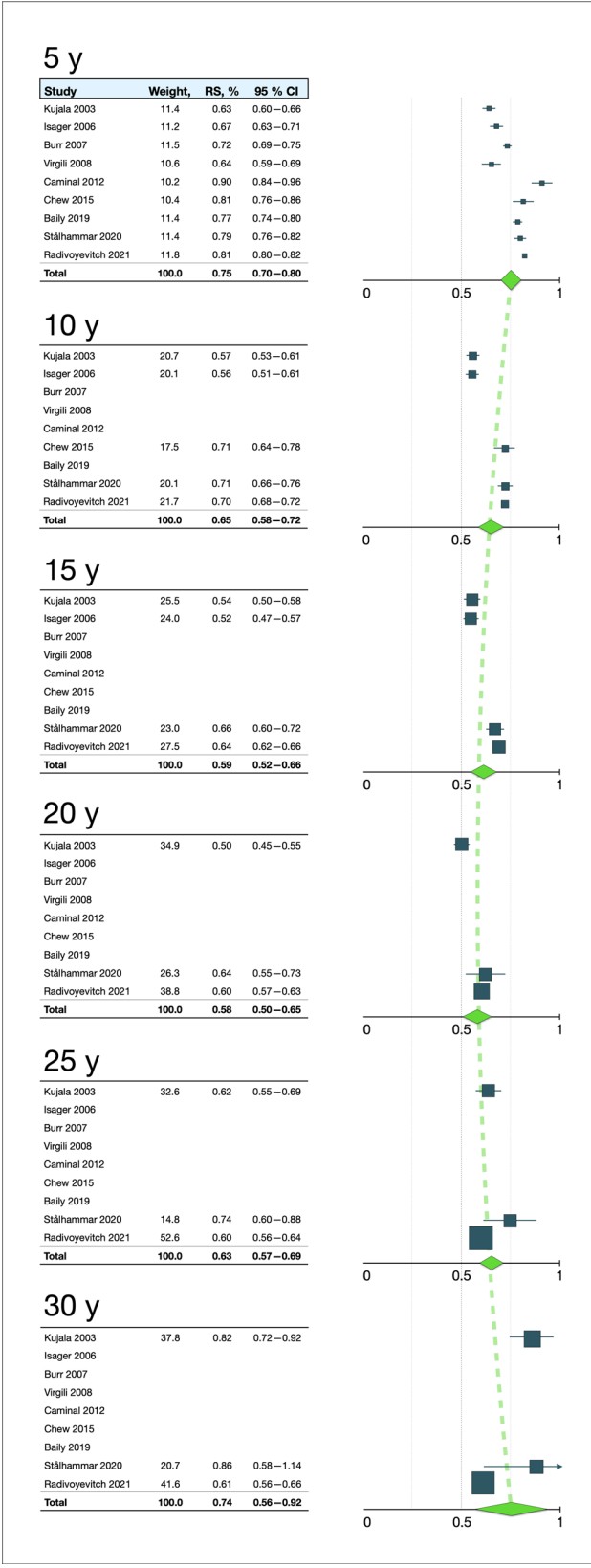

**Fig. 2 Forest plots of relative survival rates at 5-year intervals between 5 and 30 years after uveal melanoma diagnosis in a random-effects inverse-variance model (IV).** Y years. RS relative survival, CI confidence interval. Error bars represent 95% CI.

mortality reported by ourselves after plaque brachytherapy of posterior uveal melanomas with a thickness of 5.5–7.4 mm[53]. The high mortality rates in the paper by Kujala et al. could perhaps therefore be characterized as being representative of a group of patients with relatively high risk for metastasis.

Readers should bear in mind that the relative survival rates found here are estimations and averages, valid for uveal melanoma patients as a group. The presented combined estimate of mortality rates should thereby be representative of the general mortality in the disease. For any individual patient however, information on the risk of metastatic development and uveal melanoma-related death will have to be adjusted upwards or downwards based on a range of other factors, including the size and location of his or her tumor, American Joint Committee on Cancer (AJCC) stage, BAP-1 expression, loss of heterozygosity of chromosome 3, presence of vasculogenic mimicry etc[1,54–60]. Some of these factors can isolate patients with >80% risk of suffering from a melanoma-related death[58,61]. Gene expression profiling of tumor tissue samples obtained with biopsy or from enucleated specimens is used at an increasing number of institutions. It has been retrospectively and prospectively validated and shown to provide prognostic information independently of tumor size[55,62–64]. Further, several publicly available tools for prediction of metastatic probability have been developed, including The Liverpool Uveal Melanoma Prognosticator Online (LUMPO) and Predicting Risk of Metastasis in Uveal Melanoma (PriMeUM)[14,65,66]. These tools use combinations of clinical, genetic, chromosomal or histological features to arrive at accurate prognostic predictions. Depending on the outcomes of prognostic predictions, regardless of which factors these are based on, the perhaps better-than-expected survival for uveal melanoma on the group level may be of small comfort for the individual patient.

The present study has several limitations. The included articles varied in methods used for determination of uveal melanoma-related mortality. Whereas some relied on audited cause of death, others relied on collection of medical records and classifications from cancer registries. Secondly, only three of the included studies reported 20-year relative survival rates or longer, limiting the number of patients and amount of data to base results and conclusions on. Thirdly, the pooled estimates may be affected by statistical disadvantages of the used methods. In the random-effects inverse-variance model, the weight given to each study was the inverse of the variance of the relative survival rate. As we did not include studies with <100 patients and the variances were sufficiently similar, similar weights were given to all included studies. On the other hand, the weighted averages model may risk giving too much weight to studies based on sample size, disregarding smaller studies with meticulously collected data. Fourthly, methods alternative to fitting the combined estimate to data from the random-effects and weighted averages models may very well have been preferred by some, including fitting of the curve to the raw survival data from each article or to not perform curve fitting at all. This would however not have taken the sample sizes into account, and it would have given excessive consideration to deviations in the survival curve produced by single studies. Lastly, the confidence interval of our combined estimate was based on the standard deviation between the random-effects and weighted average models. This produced a quite narrow interval as the two curves were closely approximated during the first 20–25 years after diagnosis. Other methods for calculation, including taking each individual study or the cumulative range from both models into account, would likely have produced a broader confidence interval for the combined estimate. On the

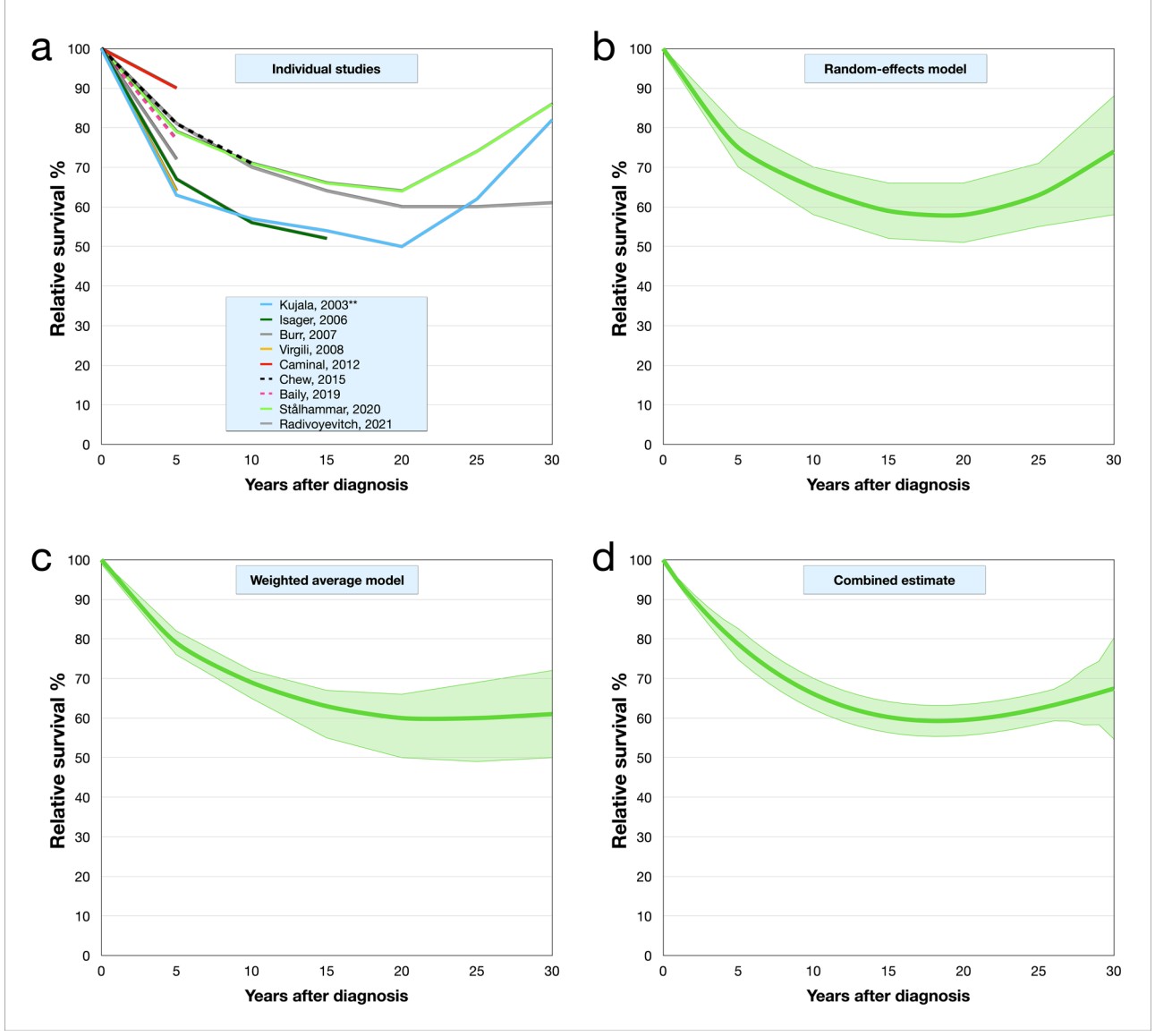

**Fig. 3 Uveal melanoma relative survival curves. a** Based on reported rates in each included study. **b** Based on meta-analysis in a random-effects inverse-variance model. **c** Based on meta-analysis in a weighted averages model. **d** Combined estimate based on curve fitting to random-effects inverse-variance and weighted averages models. Green areas represent 95% confidence intervals. The confidence interval of the combined estimate is based on the standard deviation between the random-effects and weighted average models.

other hand, previous research has shown that standard errors of relative survival of cancer patients may be substantially over-estimated with the herein used method, which may have led us to report excessively broad confidence intervals[29].

## Conclusions

About two in five patients with uveal melanoma succumb to their disease within 20 years after primary tumor treatment. Patients surviving 20 years or longer may have a survival advantage to individuals of the same sex and age from the general population. Estimations are somewhat impeded by variance in patient base-line characteristics and in methods used for data acquisition. The main source for statistical heterogeneity in this study was how-ever the large number of patients included. In only one out of three models did the confidence interval reach the often quoted 50% mortality in uveal melanoma. This may be useful when counseling patients. Future research could improve standardiza-tion of methods for reporting patient outcomes in cancer.

## Data availability

All data used in this review and meta-analysis is available from publicly available and herein referenced sources. All data generated or analyzed during this study are included in this published article and its supplementary information files. Source data used to generate Figs. 2 and 3 is provided as Supplementary Data 1 and 2, respectively.

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

## Acknowledgements
Support for this study was provided to Gustav Stålhammar from: The Swedish Cancer Society (20 0798 Fk). Region Stockholm (reference 20200356). The funding organizations had no role in the design or conduct of this research.

## Author contributions
G.S.: Conceptualization, methodology, data curation, literature search, formal analysis, project administration, software, supervision, validation, visualization, funding acquisition, writing—original draft and writing—review and editing. C.H.: Conceptualization, methodology, data curation, literature search, formal analysis, visualization, writing—original draft and writing—review and editing. Both authors directly accessed and verified the underlying data reported in the manuscript.

## Funding

## Competing interests
Gustav Stålhammar has received honoraria (August 2021) from Santen S.A., Geneva, Switzerland for writing and reviewing content and serving on a virtual advisory board in the creation of an online Ophthalmology education platform. Christina Herrspiegel declares no competing interests.
