## [Peer Review File · Communications Medicine]

Reviewers' comments:

Reviewer #1 (Remarks to the Author):

This is a well conducted study to answer an important question, relevant in considering trials in the early stage setting in uveal melanoma and also in counselling patients.

Minor comments:

Data Collection and Risk for Bias Assessment section had sentences on the authors qualifications and roles, I am not sure why this is relevant.

Major comment

There is no discussion about the molecular analysis that is increasingly possible on primary tumours and performed in several tertiary centres where primary ocular melanomas are managed. I appreciate there is significant international variation and such analysis will not be possible everywhere. Nevertheless such analysis is prognostic and some centres use it in counselling patients (eg liverpool ocular melanoma group LUMPO score). There is still quite a lot of variation in disease free survival in molecularly defined groups but it could be that the majority of relapses do occur in high risk disease eg monosomy 3. a few sentences to discuss this I think would be useful. There is debate as to if surveillance should be more intensive in the high risk group patients with de-escalation in the lower risk.

Reviewer #2 (Remarks to the Author):

This is a systematic review and meta-analysis of relative survival estimates among patients with uveal melanoma. It seems that there is substantial variation in survival rates reported, therefore this is a useful study. The systematic review and meta-analysis appears to have been conducted well and the paper is well written. I only have minor comments:

1. Although the study involves a systematic review, this is not mentioned in the title and abstract. Why is that?
2. The formula in lines 151-153 needs more explanation and a reference.
3. Line 172: what does 'less statistical bias' mean?
4. Analysis of subgroups of studies could be done to assess potential sources of heterogeneity.

Dear Reviewers,

We appreciate your insightful comments and have amended the paper to address your concerns. Please find our point-by-point response below.

Sincerely, on behalf both authors

Gustav Stålhammar
 Associate professor
 M.D. Ph.D. FEBO
 St. Erik Eye Hospital
 Eugeniavägen 12
 171 64, Stockholm
 Sweden
 Email: gustav.stalhammar@ki.se
 Phone: 0046 8672 30 00
 Fax: 0046 8672 33

Comment	Author's response	Change in the Manuscript
Editor		
Please add 'systematic review' to the manuscript title	Changed accordingly	Title page
Ensure a fully completed and updated (if page numbers change) PRISMA checklist	The checklists have been updated	Supplementary material
Reviewer #1		
Data Collection and Risk for Bias Assessment section had sentences on the authors qualifications and roles, I am not sure why this is relevant.	According to the MOOSE (Meta-analyses Of Observational Studies in Epidemiology) Checklist, the qualifications of the searchers (e.g. librarians and investigators) are to be reported. To clarify, we have added "Qualification of searchers" to the title of this section	Page 4

There is no discussion about the molecular analysis that is increasingly possible on primary tumours and performed in several tertiary centres where primary ocular melanomas are managed. I appreciate there is significant international variation and such analysis will not be possible everywhere. Nevertheless such analysis is prognostic and some centres use it in counselling patients (eg liverpool ocular melanoma group LUMPO score). There is still quite a lot of variation in disease free survival in molecularly defined groups but it could be that the majority of relapses do occur in high risk disease eg monosomy 3. a few sentences to discuss this I think would be useful. There is debate as to if surveillance should be more intensive in the high risk group patients with de-escalation in the lower risk.	Thank you for this suggestion. The third paragraph of the discussion has been amended and extended. It now reads: "Readers should bear in mind that the relative survival rates found here are estimations and averages, valid for uveal melanoma patients as a group. The presented combined estimate of mortality rates should thereby be representative of the general mortality in the disease. For any individual patient however, information on the risk of metastatic development and uveal melanoma-related death will have to be adjusted upwards or downwards based on a range of other factors, including the size and location of his or her tumor, American Joint Committee on Cancer (AJCC) stage, BAP-1 expression, loss of heterozygosity of chromosome 3 etc.^{1,51-55} Some of these factors can isolate patients with >80 % risk of suffering from a melanoma-related death.^{55,56} Gene expression profiling of tumor tissue samples obtained with biopsy or from enucleated specimens is used at an increasing number of institutions. It has been retrospectively and prospectively validated and shown to provide prognostic information independently of tumor size.^{52,57-59} Further, several publicly available tools for prediction of metastatic probability have been developed, including the The Liverpool Uveal Melanoma Prognosticator Online (LUMPO) and Predicting Risk of Metastasis in Uveal Melanoma (PriMeUM).^{14,60,61} These tools use combinations of clinical, genetic, chromosomal or histological features to arrive at accurate prognostic predictions. Depending on the outcomes of prognostic predictions, regardless of which factors these are based on, the perhaps better-than-expected survival for uveal melanoma on the group level may be of small comfort for the individual patient."	Pages 8 and 9
Reviewer #2		
Although the study involves a systematic review, this is not mentioned in the title and abstract. Why is that?	"Systematic Review" has been added to the title.	Title page
The formula in lines 151-153 needs more explanation and a reference.	We have now added that this is a common method in cancer epidemiology and provided three references. Additionally, we have also added to the limitations section of the discussion that this method may overstate the standard error and thereby lead to excessively broad confidence intervals.	Pages 4 and 9
Line 172: what does 'less statistical bias' mean?	This formulation has been removed and we now simply state that: "curve fitting was performed based on the results of the random-effects and weighted averages models to arrive at a combined estimate."	Page 5

Postal address

Department of Clinical Neuroscience, KI
SE-177 76 Stockholm
+46 8 672 30 00

gustav.stalhammar@ki.se
<https://staff.ki.se/people/gussta>

Analysis of subgroups of studies could be done to assess potential sources of heterogeneity.	We agree that this could be a good way of assess potential heterogeneity. However, after discussing it we have decided not to pursue a subgroup analysis. There are several reasons for this: Firstly, even if significant similarities or differences were found between subgroups, we would not be able to clarify them. Most of the included studies report clinical characteristics of their patients, typically including patient sex and age, and tumor size. These factors can be compared, but we will not know if differences are related to tumor location, histological, genetic, chromosomal, immunohistochemical or other factors. Secondly, subgroup analyses based on these factors would have very limited potential to reveal anything new than what is already well established (prognostically speaking). Thirdly, a meta-analysis of relative survival for patients with small, medium and large uveal melanoma, for old and young patients, men and women etc. is outside the scope of the aim of this study, and such extensive analyses may be more appropriate for a separate publication. Fourthly, an analysis of survival in relation to these classic factors would at least partially overlap previous publications. ¹⁻³	
Additional changes:	The layout of figures 1 through 3 and supplementary files have been updated The term "final estimate" has been replaced with "combined estimate" throughout the manuscript, as we feel that this is a better term for the combined result of the random-effects and weighted average models.	

References mentioned above:

1. Jouhi *et al.* The Small Fatal Choroidal Melanoma Study. A Survey by the European Ophthalmic Oncology Group. *Am J Ophthalmol.* 2019.
2. Hawkins *et al.* The Collaborative Ocular Melanoma Study (COMS) randomized trial of pre-enucleation radiation of large choroidal melanoma: IV. Ten-year mortality findings and prognostic factors. COMS report number 24. *Am J Ophthalmol.* 2004.
3. Al-Jamal *et al.* Uveal melanoma among Finnish children and young adults. *J AAPOS.* 2014.

Postal address

Department of Clinical Neuroscience, KI
SE-177 76 Stockholm
+46 8 672 30 00

gustav.stalhammar@ki.se
<https://staff.ki.se/people/gussta>

REVIEWERS' COMMENTS:

Reviewer #1 (Remarks to the Author):

The authors have addressed my comments

Reviewer #2 (Remarks to the Author):

Thank you for the revised version, my previous comments have been addressed.